# Are Gait and Balance Problems in Neurological Patients Interdependent? Enhanced Analysis Using Gait Indices, Cyclograms, Balance Parameters and Entropy

**DOI:** 10.3390/e23030359

**Published:** 2021-03-17

**Authors:** Malgorzata Syczewska, Ewa Szczerbik, Malgorzata Kalinowska, Anna Swiecicka, Grazyna Graff

**Affiliations:** Department of Rehabilitation, The Children’s Memorial Health Institute, Al. Dzieci Polskich 20, 04-730 Warszawa, Poland; e.szczerbik@ipczd.pl (E.S.); m.kalinowska@ipczd.pl (M.K.); a.swiecicka@ipczd.pl (A.S.); graffgrazyna@onet.pl (G.G.)

**Keywords:** gait, balance, neurological problems

## Abstract

Background: Balance and locomotion are two main complex functions, which require intact and efficient neuromuscular and sensory systems, and their proper integration. In many studies the assumption of their dependence is present, and some rehabilitation approaches are based on it. Other papers undermine this assumption. Therefore the aim of this study was to examine the possible dependence between gait and balance in patients with neurological or sensory integration problems, which affected their balance. Methods: 75 patients (52 with neurological diseases, 23 with sensory integration problems) participated in the study. They underwent balance assessment on Kistler force plate in two conditions, six tests on a Balance Biodex System and instrumented gait analysis with VICON. The gait and balances parameters and indices, together with entropy and cyclograms were used for the analysis. Spearman correlation, multiple regression, cluster analysis, and discriminant analysis were used as analytical tools. Results: The analysis divided patients into 2 groups with 100% correctly classified cases. Some balance and gait measures are better in the first group, but some others in the second. Conclusions: This finding confirms the hypothesis that there is no direct link between gait and balance deficits.

## 1. Introduction

Two main functions which decide about every day quality of life are: locomotion, which enables efficient transfer between various places, and balance, which ensures proper and stable body posture during different tasks. Both functions are complex, and require intact sensory systems (proprioceptive, visual, vestibular), and their proper integration by central nervous system, and efficient control signals, correcting emerging disturbances. Emerging in recent years rehabilitation practice, taking into consideration the above described features of balance and gait, and the concept of a patient as an individual interacting with environment during task performance, encourages the gait retraining in the attempt to improve the balance function of the patient [1]. This approach assumes that both functions are interdependent, using similar patient’s systems and similar pathways and neurocenters for integration of signals coming from all sensory inputs and are based on similar control rules. 

In some studies this assumption is confirmed by experimental data. In a study performed by Guffey at co-workers on healthy 2 to 4 years old children the spatio-temporal gait parameters and balance abilities were assessed. The results proved that gait parameters explained over 50% of the balance scores, indicating the dependence between the two functions [2]. The backward walking training is regarded as an efficient tool in improving balance performance, as described in a meta-analysis study by Wang and co-authors [3]. They analysed nine papers and found out that all reported beneficial effects of backward walking training on balance indices. Mudge and co-workers investigated the effect of body supported walking training on treadmill on gait and balance in patients with chronic stroke. They found out, that such training had limited influence on level walking, but significantly improved balance function [4]. Langhammer and co-workers found [5] that the results of 6 min walk test, Time-Up-And-Go test correlated with the results of Berg’s Balance Scale and Motor Assessment Scale.

In contrast to the previous studies some results of the rehabilitation procedures aimed at improving balance and gait are inconclusive. The analysis of papers dealing with efficacy and effectiveness of non-aerobic exercise program for patients suffering from traumatic brain injuries revealed, that most studies were done on small, heterogeneous sample groups, interventions were not standardized, and outcome data were of poor quality [6]. Children with intellectual disabilities often suffer from delayed motor control development and physical fitness, which manifests via balance and gait problems. Lee and co-workers investigated the effect of balance training on postural balance, gait and functional strength in a group of such patients [7]. They found out, that provided balance training improved postural balance and strength, but contrary to their expectations, no improvement of the gait function was observed. In one study concerning children with unilateral cerebral palsy only partial dependence between balance problems and gait disturbances was found: the correlations between them, although statistical significance was low [8]. Preliminary studies in neurologically and sensory impaired children, investigating the dependence of the level of gait pattern abnormalities (spatio-temporal parameters, gait indices) and balance problems (results of balance tests) did not show the dependence between the two functions [9,10]. Therefore the aim of this study was to investigate deeper the possible dependence between gait and balance in patients with neurological or sensory integration problems, which affected their balance abilities.

## 2. Materials and Methods

### 2.1. Material

Seventy five patients participated in the study, 52 with diagnoses of neurological diseases (cerebral palsy, Guillain-Barre syndrome, polyneuropathy, traumatic brain injury, etc.), and 23 with sensory integration problems, affecting mainly balance abilities. The group consisted of 38 boys and 37 girls aged from 5 to 17 years old. BMI index was within normal range (taking into consideration age and sex) in all but one patient, who was over weighted. The study was approved by the Local Bioethical Committee, parents of all children and adolescents gave their informed consent, as well as all participant who were 16 or older.

### 2.2. Methods

#### 2.2.1. Balance

Balance assessment was performed using two types of equipment: Kistler force plates and a Biodex Balance System (BBS).

##### Balance on Kistler Force Plate

The balance was evaluated in two conditions: with eyes open and with eyes closed. In both conditions the patient was asked to stand still, with both feet parallel to each other, with distance between them approximately equal to the patient’s pelvis width. The arms were hanging freely, and during the standing with eyes open he/she was asked to look straight ahead, and with eyes closed to keep similar position of the head. The data was collected at the sampling frequency of 1000 Hz, and the trials’ time was 30 sec. During analysis the data were resampled to 60 Hz. The following parameters were calculated from the data: maximal radius of the sway, average radius of the sway, maximal forward sway, maximal backward sway, maximal sway to the left, maximal sway to the right, total length of the sway path. The number of parameters exceeding the normal values [11] were also noted, separately for the eyes open/eyes closed condition. Additionally the Shannon entropy of the sway path was calculated from the X, Y matrix using Matlab (R2013a) procedure, after normalization of the data to the maximal values of the matrix, separately for the eyes open and eyes closed condition.

##### Balance on Biodex Balance System SD

Patients underwent several balance tests on a Biodex Balance System SD. Each test was repeated three times, and each repetition lasted 20 sec. There was 10 sec break between repetitions. The platform’s system calculated a set of indices, and the overall balance index from each test was taken for the further analysis. The patients were situated on the platform according to the requirements described in the BBS manual [12]. The following tests were performed:-On a stable platform with eyes open;-On an unstable platform with eyes open (level of stability 4);-On a slightly unstable platform with eyes open (level of stability 8);-On a platform with changing level of stability with eyes open (from level 12 to 4);-Limits of Stability Test (LOS) with moderate difficulty level [12] on stable platform;-Modified Clinical Test of Sensory Integration in Balance (m-CTSIB). This test comprised four different conditions: standing on firm surface with eyes open, standing on firm surface with eyes closed, standing on foam with eyes open, and standing on foam with eyes closed.

#### 2.2.2. Gait Analysis

The patients underwent also the objective, instrumented gait analysis, which was performed using 12 T40-S camera VICON MX system. The gait analysis was done the same day as the balance tests. The lower body Plug-In-Gait marker set and model were used. The patients walked with their self-selected gait speed, after they got acquainted with the environment, equipment and tasks. Later six trials were captured using the Nexus software. Part of the processing of the captured trials was calculation of the Gait Deviation Index (GDI), which was later averaged for trials and legs. GDI was calculated using the Nexus software. The kinematics from six trials was later averaged in the Polygon software and expressed as per cent of the gait cycle. The averaged data was exported with values representing every 2% of the gait cycle. From the averaged kinematic data the Gait Variables Scores (GVS) were calculated for left and right leg separately [13]. The Gait Variable Score (GVS) is calculated as root mean square (RMS) difference between kinematic variable across the gait cycle of the patient and reference variable representing healthy subjects. The GVS are calculated for nine key kinematic variables: pelvic tilt, hip flexion, knee flexion, ankle dorsi/plantar flexion, pelvic obliquity, hip ab/adduction, pelvic rotation, and foot progression, and from them a Gait Profile Score (GPS) can be calculated [13]. The calculations were performed with our own procedures, using the lab’s reference kinematics data.

The exported averaged kinematic data were also used to calculated the cyclogram index. Cyclogram, called also an angle-angle diagram, reflects the coordination abilities of the patient during gait. On one axis there are values of an angle in one joint throughout the gait cycle, on the second one values of the second joint, both angles in the same plane. According to Goswami cyclograms reflect the kinematic changes within the whole gait cycle and the coupling between the joints [14]. In the present study the following cyclograms were created for each patient, separately for the left and right leg: hip-pelvis in sagittal plane, knee-hip in sagittal plane, ankle-knee in sagittal plane, and hip-pelvis in frontal plane. For each cyclogram its perimeter was calculated, as one of the geometric characteristics of the closed-loop cyclogram. The patient’s perimeter of each cyclogram was normalized to the perimeter of the respective normative cyclogram (from the lab’s reference kinematics data). For each patient a cyclogram index was calculated as an average of the normalized perimeters of the above described cyclograms, separately for left and right leg, and later as one averaged cyclogram index. Figure 1a presents the reference cyclograms of healthy reference data, and 1b an example of the cyclograms of one of the patients.

#### 2.2.3. Statistical Analysis

All calculations were performed using Statistica software v10.0 (StatSoft Inc. (now part of TIBCO Software Inc., Palo Alto, CA, USA, the cut-off p-level was set to 0.05. The following tests were used for the analyses: rank Spearman correlation, multiple regression, cluster analysis using connectivity-based clustering with weighted group method with medians (averaged linkage clustering) and no assumption of the number of groups, and finally the discriminant analysis. 

Spearman rank correlation test was used to establish the dependence between entropy in eyes open and eyes closed conditions, and to see if there is a link between standard method of balance assessment (balance parameters and indices) and entropy. Multiple regression method was used to check the dependence between gait and balance parameters. Two groups of these analyses were performed: first when entropy and balance indices were dependent variables, and gait indices were independent variables, and second when gait indices were dependent variables, and balance indices independent ones. Cluster analysis was performed to see if patients with neurological and sensory integration problems could be grouped based on their gait and balance parameters. The averaged linkage clustering method was used. In this method the distance between the cases is calculated and the criterion for linking them is the smallest distance. Later the basic clusters are hierarchically link with each other to form the cluster tree. Based on the cluster analysis, i.e. visual inspection of the hierarchical cluster tree, the patients were divided into two groups, and discriminant analysis was used to reveal which parameters were the predictors deciding to which of these two groups a patient was assigned. The division of the patients into two groups was arbitral (a criterion used for division was to end up with smallest number of groups, in which the distances (showed by the clustering loops) were relatively small. As there was no golden standard to which the division can be compared, we used the discriminant analysis as an internal evaluation of the quality of the cluster analysis. Additionally the t-Student test was performed to see if the demographical parameters differ between the two groups. 

## 3. Results

There is statistically significant correlation (moderate) between entropy in eyes closed and eyes open condition: R = 0.469. Other statistically significant correlations found during analysis were in eyes open condition: between entropy and number of parameters exceeding normal values (Kistler balance test) (R = 0.675), between entropy and total sway path (R = 0.614); in eyes closed condition: between entropy and number of parameters exceeding normal values (Kistler balance test) (R = 0.683), between entropy and total sway path (R = 0.640). The results of the multiple regression tests are presented in Table 1.

Figure 2 presents the hierarchical clustering graph, on which the division of the patients into two groups was done. 

The results of the discriminant analysis, which was done based on this division, are presented in Table 2 and Table 3. 

Table 4 presents the summary statistics of all analyzed parameters for all patients, and groups 1 and 2, which arisen from the clustering, and Table 5 their demographic parameters. 

## 4. Discussion

The aim of this study was to examine the possible dependence of the gait pattern and balance problems in patients with neurological problems and balance deficits, as there are contradictory statements in the literature. To better assess if such a dependence exists we used balance parameters from two balance tests performed on the Kistler force plate (eyes open, and eyes closed), and six balance tests performed on the Biodex Balance System. The set of these tests reflected different balance situation, which patients could encountered in the daily life: standing on stable surface, on unstable surface with different stability levels, foam, etc. In case of quiet standing tests performed on Kistler platform apart from most popular parameters used to describe balance ability (radius of sway, total sway path, etc.) the entropy of the sway path was calculated for both tests as a measure of complexity and regularity of the paths. In case of the gait evaluation we used commonly used indices assessing the overall gait patterns such as GDI, GPS and GVS, as well as additional measures: cyclograms and calculated from them cyclogram index, which reflect the inter-joint coordination ability of the patients. The analyses of the data confirmed our preliminary findings [9,10] that there is no direct link between the patient’s gait pattern and his/her balance dysfunctions. In the first study the data of 63 patients were used. The multiple regression was used to find out if there is any dependence between balance tests performed on Biodex Balance System (indices provided from the Biodex software) and Kistler force plate (total path length) and GDI and GVS. In the second the data of 47 patients with neurological problems and sensory integration problems were used. The aim of the study was to see if the type of the problem (i.e. neurological disease) or the results on the Biodex Balance System show any dependence on gait. Patients underwent two balance tests: on stable and unstable platform, their results were expressed as 0 –normal, 1 - abnormal, with eyes open, and from the results of gait analysis GDI, GVS and GPS were calculated. The logistic regression and discriminant analysis were used to look for predictive parameters. In both studies no direct dependence between gait dysfunction and balance problems were found, thus the present study was designed to look more deeply into the problem of possible dependence by increasing the number of patients, addition of balance parameters, nonlinear measures, and cyclograms, as well as applying the cluster analysis to the data.

Multiple regressions revealed some dependences between balance and gait parameters, but in each of these analyses only few parameters were indicated as statistically significant: from one to four. Moreover, there is no consistency between the results: different gait parameters were indicated as statistically significant for different balance parameters and vice versa (see Table 1), except for overall gait index GDI (mostly based on kinematics in sagittal plane), entropy in eyes open condition and Biodex stability index during standing on stable platform. 

To check if the conclusion about lack of direct dependence between gait and balance, arising from the results of multiple regression was correct, a cluster analysis of the patients based on the Euclidean distance between them was performed. All gait and balance parameters were used for the clustering (Figure 2) of cases. The clustering enabled the division of the patients into two groups (based on the visual inspection of the clustering tree, Figure 1), and for those groups discriminant analysis was performed to see which of the balance and gait parameters were predictors, deciding to which groups the patient was assigned. The analysis showed that statistically significant predictors were both balance parameters from Kistler and Biodex tests, and gait indices (see Table 2). Surprising result was perfect classification of all the patients done by the discriminant model to the groups arisen from the clustering: 100% of cases were correctly classified (see Table 3).

The summary statistics of all parameters (Table 4) calculated for the whole group of patients, as well as separately for the groups arisen from the clustering, revealed, that patients classified to the group no 1 had higher values of most of the balance and gait parameters than patients from the group no 2, except for Biodex stability index on mildly unstable platform, Biodex stability index for LOS test, GPS (Gait Profile Score), and following Gait Variable Scores (GVS): pelvis, knee and ankle in sagittal plane pelvis in frontal plane, hip in frontal plane, hip in transversal plane. These results also point to the fact, that there is no direct link between the pathological gait pattern and balance deficits, as some balance and gait parameters were higher in group no 2, while all parameters from Kistler test (including entropy), four Biodex indices, all cyclograms and Gait Deviation Index (GDI), GVS hip sagittal plane and GVS foot progression were lower in this group The higher values of the parameters meant, that the results were worse, while the lower values of the parameters, indicated that they were better.

The posture control develops in children from birth till the adulthood, with most intensive development till the 6th year of age [15,16], and starts with the head control. The improvement of the postural control occurring with age is accompanied by gradual improvement of control over many degrees of freedom which should be controlled simultaneously during many coordinated functional tasks. 

Practically in all creatures rhythmic and stereotyped motor tasks, like locomotion, swimming and flying are controlled to a large extend by spinal central pattern generators, which, in vertebrates, are capable under specific conditions, to produce rhythmic and coordinated movements, even in absence of descending of peripheral inputs [17]. Proper posture and balance control are required for efficient and independent gait, but walking is also controlled by the locomotor pattern generator. Moreover, the balance tests done in standing positions, represent static balance, while walking is a dynamic task, with periodical loss and retrieve of balance. These are probably the reasons behind our findings: although our patients suffered from balance deficits, and had abnormal gait patterns we could not establish the direct link between the level of gait pathology and balance deficits. The introduction to our study the seldom used balance (entropy) and gait measures (cyclograms) as additional features assessing the complexity, regularity and inter-joint coordination abilities, confirmed our previous findings of lack of direct dependence between gait and balance.

One of the limitations of the study is the unbalance number of patients in the groups which arisen from the cluster analysis. This was not intended: the analysis was performed after the recruitment process was finished and all the data were collected. Maybe the proportion (approximately) 1:3.5 reflects differences between neurological patients concerning gait and balance deficits, but more studies to confirm this finding are needed.

## Figures and Tables

**Figure 1 entropy-23-00359-f001:**
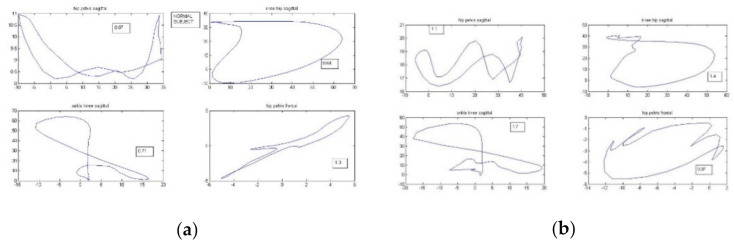
Cyclograms: (**a**) for a healthy subject, (**b**) exemplary one from one of the patients. The numbers in squares are perimeters of each cyclogram. The upper left graph represents cyclogram of hip-pelvis in sagittal plane, upper right: knee-hip in sagittal plane, lower left: ankle-knee in sagittal plane, and lower right the cyclogram of hip-pelvis in frontal plane.

**Figure 2 entropy-23-00359-f002:**
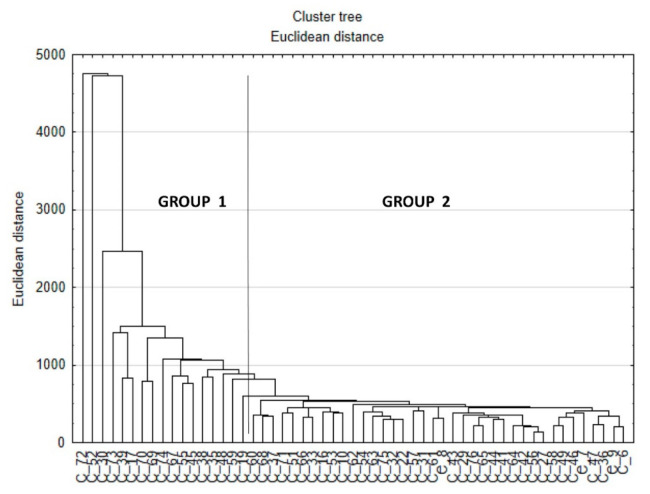
The hierarchical graph of the cluster analysis. X axis—cases, Y-axis—Euclidean distance on which order of clustering was done. The vertical line shows the division into two groups.

**Table 1 entropy-23-00359-t001:** The results of the multiple regression analyses.

Dependent Variable	F Test	Statistically Significant Independent Variables
Entropy eyes open (Kistler)	F = 3.509, *p* = 0.007	Ankle-knee cyclogram sagittal plane, GVS hip frontal plane, GDI, knee-hip cyclogram sagittal plane
Entropy eyes closed (Kistler)	F = 5.054, *p* < 0.001	GVS hip transversal plane, GVS pelvis transversal plane, GVS hip sagittal plane, GVS knee sagittal plane
Stability index on stable platform (Biodex)	F = 8.217, *p* < 0.001	GDI, cyclogram index
Stability index on mildly unstable platform (Biodex)	F = 3.755, *p* = 0.005	Knee-hip cyclogram sagittal plane, GVS hip transversal plane
Stability index on unstable platform (Biodex)	F = 4.148, *p* = 0.004	Knee-hip cyclogram sagittal plane, cyclogram index, GVS hip transversal plane
Limits of Stability index (Biodex)	F = 7.180, *p* < 0.001	GDI, GVS knee sagittal plane
mCTSIB	F = 4.202, *p* = 0.009	GDI
GPS	F = 8.110, *p* < 0.001	Stability index on stable platform (Biodex), total length path eyes closed, maximal sway to the right eyes closed,
GDI	F = 7.693, *p* < 0.001	Stability index on stable platform (Biodex)

**Table 2 entropy-23-00359-t002:** The gait and balance parameters included into the model of the discriminant analysis, which were statistically significant (F = 12.490, *p* < 0.001). The parameters which appeared to be discriminant are marked with bolded text and asterix.

	F to Remove (1, 40)	*p*
No of parameters exceeding normal values in eyes closed condition (Kistler)	13.582	**<0.001 ***
Entropy eyes open (Kistler)	1.037	0.315
Maximal sway to the right in eyes closed condition (Kistler)	5.441	**0.025 ***
Hip-pelvis cyclogram in sagittal plane	5.204	**0.028 ***
GVS ankle sagittal plane	12.157	**0.001 ***
Maximal radius of sway in eyes closed condition (Kistler)	7.377	**0.010 ***
Average radius of sway in eyes closed condition (Kistler)	9.320	**0.001 ***
Entropy eyes closed (Kistler)	11.953	**0.001 ***
Total path length in eyes open condition (Kistler)	9.480	**0.004 ***
GVS knee sagittal plane	7.824	**0.008 ***
GVS hip sagittal plane	11.039	**0.002 ***
Entropy eyes closed (Kistler)	9.251	**0.004 ***
Stability index on unstable platform (Biodex)	5.654	**0.022 ***
Stability index on platform with changing stability (Biodex)	8.620	**0.005 ***
GVS pelvis transversal plane	0.062	0.805
Stability index in mCSTIB test	4.219	**0.047 ***
Hip-pelvis cyclogram in frontal plane	8.479	**0.006 ***
GVS hip transversal plane	7.702	**0.008 ***
Maximal sway to the left in eyes closed condition (Kistler)	7.482	**0.009 ***
GDI	4.617	**0.038 ***
GVS pelvis sagittal plane	2.714	0.148

**Table 3 entropy-23-00359-t003:** The classification matrix for the discriminant analysis model.

	Correctly Classified Cases [%]	Group 1	Group 2
Group 1	100	17	0
Group 2	100	0	44
Total	100	17	44

**Table 4 entropy-23-00359-t004:** The parameters analyzed in study, summarized by medians and quartiles: for all the patients and for the groups which arisen from cluster analysis.

Parameters	All Patients	Group 1	Group 2
	Kistler Eyes	Open Condition	
Maximal radius of sway	18.65 <13.75–24.05>	24.3 <20.2–35.8>	18.1 <13.2–22.57>
Average radius of sway	6.4 <4.75–9.25>	8.5 <6.0–10.9>	6.2 <4.5–8.2>
Total sway path	513.0 <386.5–634.5>	690.0 <562.0–853.0>	481.0 <345.0–557.0>
Maximal sway to the left	11.0 <8.35–15.95>	15.6 <9.4–23.6>	10.9 <8.1–14.2>
Maximal sway to the right	11.4 <8.75–18.6>	17.6 <11.5–24.4>	10.6 <7.5–15.0>
Maximal forward sway	13.3 <9.75–18.35>	17.8 <13.8–23.5>	12.3 <9.5–17.1 >
Maximal backward sway	11.4 <10.7–19.7>	19.0 <16.1–20.6>	13.6 <10.3–18.0>
No of parameters exceeding normal values	1 <0–1.5>	2 <1–3>	1 <0–1>
Entropy	2892.8 <2159.5–3734.6>	3932.9 <2894.4–4967.9>	2832.7 <2058.6–3308.4>
	**Kistler Eyes**	**Closed Condition**	
Maximal radius of sway	21.95 <18.05–30.5>	34.5 <26.1–52.9>	20.6 <15.6–26.6>
Average radius of sway	7.8 <6.05–10.65>	11.6 <8.4–15.7>	7.2 <6.0–9.4>
Total sway path	641.6 <469.0–977.5>	1136.0 <880.0–1481.0>	591.0 <439.0–765.0>
Maximal sway to the left	15.4 <10.55–20.0>	19.7 <18.0–31.0>	13.4 <9.99–18.8>
Maximal sway to the right	14.55 <9.7–21.45>	17.0 <13.9–40.3>	12.9 <9.0–19.4>
Maximal forward sway	18.0 <11.75–26.3>	31.4 <19.6–39.5>	16.0 <11.0–21.8>
Maximal backward sway	17.0 <13.15–23.05>	28.9 <22.5–31.0>	15.1 <11.7–20.5>
No of parameters exceeding normal values	1 <0–2>	3 <1–5>	1 <0–1>
Entropy	3287.05 <2570.05–4736.1>	5112.4 <4538.6–6560.5>	2962.2 <2362.3 -3753.9>
	**Biodex**		
Stability index on stable platform	0.7 <0.5–1.1>	1.1 <0.6–1.2>	0.6 <0.4–1.1>
Stability index on slightly unstable platform	0.8 <0.6–1.1>	0.7 <0.6–1.0>	0.8 <0.6–1.1>
Stability index on unstable platform	0.95 <0.7–1.4>	1.0 <0.7–1.4>	0.9 <0.6–1.3>
Stability index on platform with changing instability	0.95 <0.7–1.4>	0.9 <0.6–1.4>	0.9 <0.7–1.3>
Stability index of LOS test	38.0 <27.0–55.0>	27.0 <17.0–42>	42.5 <31.0–57.5>
Stability index of mCSTIB test	2.22 <1.58–2.83>	2.94 <2.16–4.0>	2.115 <1.53–2.58>
	**Gait**		
GDI	82.1 <73.45–87.0>	83.15 <73.45–87.9>	81.95 <73.45–86.95>
GPS	7.15 <6.23–8.14>	6.78 <6.2–7.95>	7.22 <6.26–8.26>
GVS pelvis sagittal plane	4.02 <2.25–6.77>	3.20 <2.02–7.41>	4.08 <2.44–6.2>
GVS hip sagittal plane	8.13 <6.24–10.85>	8.8 <6.18–10.38>	8.13 <6.51–11.07>
GVS knee sagittal plane	11.94 <9.84–13.4>	11.15 <9.88–12.61>	12.29 <9.72–13.71>
GVS ankle sagittal plane	6.64 <5.36–7.91>	6.0 <4.83–7.91>	6.67 <5.48–7.91>
GVS pelvis frontal plane	2.56 <1.84–3.28>	1.92 <1.55–3.73>	2.61 <2.16–3.19>
GVS hip frontal plane	3.5 <2.69–4.74>	3.37 <2.69–4.8>	3.62 <2.69–4.68>
GVS pelvis transversal plane	4.09 <2.65–5.83>	4.13 <2.62–4.84>	4.05 <2.78–5.99>
GVS hip transversal plane	11.65 <9.51–15.03>	11.37 <10.09–14.54>	11.79 <9.11–15.99>
GVS foot progression	8.27 <5.84–10.69>	8.8 <5.35–10.77>	8.21 <5.96–10.61>
Hip-pelvis cyclogram sagittal	−0.011 <−0.105–0.099>	−0.006 <−0.119–0.107>	−0.024 <−0.097–0.09>
Knee-hip cyclogram sagittal	−0.121 <−0.20–−0.037>	−0.148 <−0.206–−0.017>	−0.12 <−0.2–−0.037>
Ankle-hip cyclogram sagittal	−0.053 <−0.152–0.053>	−0.084 <−0.192–0.051>	−0.035 <−0.148–0.054>
Hip-pelvis cyclogram frontal	0.064 <−0.125–0.218>	0.077 <−0.124–0.22>	0.056 <−0.125–0.216>
Cyclogram index	−0.028 <−0.108–0.068>	−0.003 <−0.133–0.052>	−0.032 <−0.104–0.072>

**Table 5 entropy-23-00359-t005:** Demographic characteristics of the patients, summarized by medians and standard deviations. No statistically significant differences were found between the groups (t-Student test was performed).

	All Patients	Group no 1	Group no 2
**Age (years)**	11.4 ± 3.5	10.8 ± 3.5	11.6 ± 3.5
**Height (cm)**	149.4 ± 20.3	146.3 ± 15.5	150.2 ± 21.6
**Weight (kg)**	48.8 ± 22.7	40.6 ± 14.4	51.2 ± 24.1
**BMI**	19.4 ± 5.1	18.4 ± 3.7	19.6 ± 5.4
**Females/Males**	35/40	7/10	28/30

## Data Availability

The data presented in this study are available on request from the corresponding author. The data are not publicly available due to the conditions of the agreement granted by the Local Bioethical Committee and internal regulations.

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
