# Peer review of "Are Gait and Balance Problems in Neurological Patients Interdependent? Enhanced Analysis Using Gait Indices, Cyclograms, Balance Parameters and Entropy"

_entropy, 2021, doi:10.3390/e23030359_

Round 1
Reviewer 1 Report
This paper aims to investigate the dependence between gait and balance in neurological patients. Answering this question could help developing more useful rehabilitation solutions for these patients. The study uses data collected from 52 patients (5-17 years of age) with neurological disease and 23 (5-17 years of age) with sensory integration problems. The data consisted of balance and gait performance.
The organization of the paper needs to be improved. There is no details on how the clustering analysis has been performed and the F score has been calculated. The method section needs to have all the details about the clustering and F score calculations. More details on the analysis performed in the paper need to be provided. Not sure how the classification was performed and what it means. Also, there need to be healthy control subjects in this study. Another problem that needs to be addressed, the imbalanced number of subjects in each group. The authors are encouraged to provide a detailed table with subject demographics. Overall, the work is interesting; however, a significant amount of detail is missing, which doesn’t let me provide more useful feedback.
Reviewer 2 Report
An objective of this manuscript was to revise the possible dependence between gait and balance in patients with neurological or sensory integration problems. The patient balance assessment on Kistler force plate in two conditions, six tests on Balance Biodex System and instrumented gait analysis have been performed with VICON. The gait and balances parameters and indices, together with entropy and cyclograms have been used for the analysis.
The paper has been written in well done form and confirmed the result: there is no direct link between gait and balance deficits using following statistical techniques: Spearman correlation, multiple regression, cluster analysis, and discriminant analysis.
Spearman rank correlation test was used to establish the dependence between entropy in eyes open and eyes closed conditions, and to see if there is a link between standard method of balance assessment (balance parameters and indices) and entropy. Multiple regression method was used to check the dependence between gait and balance parameters.
An objective of this manuscript was to revise the possible dependence between gait and balance in patients with neurological or sensory integration problems. The patient balance assessment on Kistler force plate in two conditions, six tests on Balance Biodex System and instrumented gait analysis have been performed with VICON. The gait and balances parameters and indices, together with entropy and cyclograms have been used for the analysis.
The paper has been written in well done form and confirmed the result: there is no direct link between gait and balance deficits using following statistical techniques: Spearman correlation, multiple regression, cluster analysis, and discriminant analysis.
Spearman rank correlation test was used to establish the dependence between entropy in eyes open and eyes closed conditions, and to see if there is a link between standard method of balance assessment (balance parameters and indices) and entropy. Multiple regression method was used to check the dependence between gait and balance parameters.
This reviewer thinks that authors should provide more discussions about their previous studies that they mentioned papers [9, 10].
Please present the comparison of this paper results with your previous papers that devoted to the same problem. You should justify that there are no significant intersections in the experimental results performed and in the conclusions of these experiments.
Round 2
Reviewer 1 Report
The authors addressed my comments.